# Identification and characterization of tryptophan metabolism-related genes in carotid artery plaques

Xiaodong Jia[1☯], Zhen Wang[2☯], Lin Bai[3☯], Jianmei Wei[1], Xizi Wang[1], Zhaona Song[1], Xianchao Guo[3], Runze Jiang[3*], Qiang Zhang[1*]

**1** Joint Laboratory for Translational Medicine Research, Liaocheng People's Hospital, Liaocheng, Shandong, China, **2** Neck- Shoulder and Lumbocrural Pain Hospital, Shandong First Medical University, Jinan, Shandong, China, **3** Traditional Chinese Medicine Innovation Research Institute, Shandong University of Traditional Chinese Medicine, Jinan, Shandong, China

☯ These authors contributed equally to this work
* qiangzhang72@126.com; solon.jiang@gmail.com

## Abstract

Cardiovascular and cerebrovascular diseases, often caused by atherosclerosis, are the stern cause of death worldwide, and carotid plaque play a crucial role in the development of these diseases. Tryptophan metabolism is an important pathway involved in the regulation of immune response, inflammation and vascular health. In this study, we analyzed bulk and scRNA data from carotid plaque to investigate the relevance between tryptophan metabolism and plaque formation. We identified 446 differentially expressed genes that are enriched in immune and tryptophan-related pathways. Focusing on tryptophan metabolism, we identified six key tryptophan-related differentially expressed genes: TPH1, MAOB, TDO2, KMO, KYNU, and CYP1B1. Using the six genes, we constructed a logistic regression model with an AUC of 0.75, which successfully predicted the risk of carotid plaque formation. Analysis of global and single-cell data revealed differential expression patterns and related modes of action of the six genes in carotid plaque, suggesting that they influence the development of carotid plaque through their involvement in tryptophan metabolism, lipid biosynthesis, and inflammatory responses. These tryptophan-related differentially expressed genes can be used as potential biomarkers to assess plaque risk and as therapeutic targets to manage carotid atherosclerosis by regulating tryptophan metabolism and reducing inflammation.

## Introduction

Cardiovascular and cerebrovascular diseases are among the major causes of death worldwide, with a mortality rate as high as 20% [1]. Cardiovascular and cerebrovascular diseases refer to disorders that affect the blood vessels of the heart and brain, including coronary heart disease, stroke, hypertension, etc. They are typically caused

**Data availability statement:** The data that support the findings of this study are available from the datasets GSE43292 and GSE159677 which available in the NCBI GEO (https://www.ncbi.nlm.nih.gov/geo/) public repositoriy. All the analysis scripts used in this study have been stored on GitHub (https://github.com/Bailey0418/Carotid.plaque).

**Funding:** The author(s) received no specific funding for this work.

**Competing interests:** The authors have declared that no competing interests exist.

by atherosclerosis, which leads to narrowing, obstruction, or rupture of blood vessels and can be fatal in severe cases [2]. Atherosclerotic vascular diseases range from thickening of the innermost layer of the artery, the intima, to more complex focal structures protruding into the arterial lumen, namely atherosclerotic plaques [3]. Carotid plaques refer to the deposition of lipids, fibrous tissue, calcium deposits, and other substances on the intimal surface of the carotid artery. Their formation mechanism is mainly related to some risk factors, such as hypertension, hyperlipidemia, smoking, diabetes, and unhealthy lifestyles. These factors can cause damage to the arterial intima, subsequently leading to the accumulation of cholesterol and other lipids within the vessel wall and the formation of plaques. Over time, plaques may become unstable and rupture, forming thrombi, thereby triggering acute cerebrovascular events. Furthermore, the calcification of plaques can result in vessel stiffness and reduced elasticity, further increasing the risk of cardiovascular and cerebrovascular events [4].

Tryptophan is an essential amino acid that the human body cannot synthesize and must be obtained through diet. Beyond serving as a building block for proteins, tryptophan participates in neurotransmitter synthesis, immune regulation, and the modulation of sleep and mood. Its levels depend both on dietary intake and the activity of three main metabolic pathways: the kynurenine (Kyn), 5-hydroxytryptamine (5-HT), and indole pathways [5].The kynurenine pathway, responsible for over 95% of tryptophan degradation, is regulated by the rate-limiting enzymes TDO (Tryptophan 2,3-dioxygenase), IDO1 (Indoleamine 2,3-dioxygenase 1), and IDO2 (Indoleamine 2,3-dioxygenase 2) [6]. This pathway generates kynurenine and related metabolites that modulate inflammation, immune responses, and excitatory neural signaling, and have been linked to various diseases. Alterations in kynurenine pathway activity are associated with carotid plaque formation and instability: Munn et al. reported that changes in key enzyme activity correlate with plaque instability [7], while Shen et al. demonstrated that IDO1 influences atherosclerosis via local inflammation and immune regulation [8]. Kynurenine metabolites may also affect endothelial function, further contributing to plaque development. The 5-HT pathway converts tryptophan into 5-hydroxytryptophan (5-HTP) and subsequently into 5-hydroxytryptamine (serotonin) [9]. While 5-HT functions primarily as a neurotransmitter, it also impacts vascular endothelium and platelets, potentially promoting plaque formation through regulation of vascular tone, smooth muscle proliferation, and platelet activation. The precise contribution of serotonin to carotid atherosclerosis remains under investigation. Finally, the indole pathway, mediated by gut microbiota, produces indole and its derivatives, which help maintain intestinal homeostasis, modulate systemic inflammation, and regulate immune responses [10]. Given that atherosclerosis is part of the systemic inflammatory response, indole metabolites may influence plaque formation and stability by reducing inflammation and protecting vascular endothelium [11,12].

Research on the relationship between tryptophan metabolism and carotid artery plaque formation is crucial for understanding the underlying mechanisms of atherosclerosis, particularly how different metabolic pathways like the kynurenine, serotonin, and indole routes contribute to plaque stability and progression. This

knowledge could provide new therapeutic targets for preventing and treating carotid atherosclerosis by modulating specific tryptophan metabolic pathways, thereby reducing inflammation and improving vascular health.

## Materials and methods

### Data resource

We obtained bulk data (GSE43292) and single-cell transcriptomics data (GSE159677) for carotid plaque from the Gene Expression Omnibus (GEO) database. GSE43292 includes 32 patients, each providing a plaque sample from the carotid artery and a macroscopically intact tissue sample from a distant site. GSE159677 includes three patients, each providing an atherosclerotic core plaque sample from carotid endarterectomy and a matched tissue sample from the proximal carotid region. The gene expression levels were log-transformed.

### Differential expression and gene set enrichment analysis

The differentially expressed genes (DEGs) were identified by the R package 'limma' to compare carotid plaque and normal tissue. DEGs were defined as genes with an $p.value < 0.05$ and $|log2FC| > 0.75$ (FC, fold change). The p values were calculated using the Benjamini-Hochberg (BH) correction method.

For gene set enrichment analysis, the R package 'clusterProfiler' [13] was employed to analyze DEGs for enrichment in Gene Ontology (GO) terms and Kyoto Encyclopedia of Genes and Genomes (KEGG) pathways, with a significance threshold of $q.value < 0.05$.

### Construction of logistic regression model

A logistic regression model was constructed using the tryptophan-related DEGs (TRDEGs), with gene expression levels as independent variables and plaque occurrence as the dependent variable. The 'glm' function in R was used to build the binary logistic regression model. During the model training process, the dataset was randomly split into training and test sets in an 80:20 ratio using the 'caTools' package. The logistic regression model was trained on the training data, and coefficient estimates were analyzed. To assess model performance, the area under the ROC curve (AUC) was calculated, and confusion matrix validation was performed using the 'caret' package. Additionally, the regression coefficients and their confidence intervals were extracted, and the coefficients were cleaned and formatted using the 'broom' package. The exponentiated coefficients were computed to interpret the impact of each independent variable on plaque risk. For the test set, the model generated predicted probabilities, which were converted into binary classification results using a threshold of 0.5. To visualize the model's performance, a probability distribution plot was created using the 'ggplot2' package, and a coefficient plot was generated using the 'dotwhisker' package.

To verify the robustness of the six-gene signature, we trained four supervised classifiers, including logistic regression (GLM), LASSO-regularized logistic regression, Random Forest, and support vector machine (SVM, radial kernel). All models were trained using the same six TRDEGs as predictors, and evaluated by stratified 5-fold cross-validation to avoid sampling bias. Model performance was assessed using (1) area under the ROC curve (AUC) for discrimination, (2) area under the precision–recall curve (PR-AUC) for performance under class imbalance, and (3) the Brier score to quantify probability calibration. All classifiers were implemented using the caret framework in R, ensuring identical data partitioning and evaluation procedures across models. ROC and PR curves were generated for visualization, and a summary performance table was created to compare classifier outputs.

### Development of PPI network for TRDEGs

Using the STRING [14] database, a protein-protein interaction (PPI) network was constructed based on the TRDEGs. Interaction relationships and functional annotation information for the TRDEGs were extracted from the database. Using

the interaction data obtained from STRING, degree centrality, betweenness centrality, and closeness centrality for each TRDEG were calculated utilizing the R package 'igraph'.

## Evaluation of cell infiltration proportions of carotid plaques microenvironment

To investigate the immune cell composition in carotid artery plaques, we employed the CIBERSORT [15] algorithm. The expression data were transformed by taking the anti-logarithm, followed by reformatting with gene symbols as row identifiers. We utilized the LM22 signature matrix, which profiles 22 immune cell subtypes, to perform CIBERSORT analysis. The analysis was run with 1,000 permutations and quantile normalization enabled to ensure robust results.

## Single-cell RNA-seq data analysis

The R package 'Seurat' was used to process single-cell RNA sequencing data. In the initial quality control, the mitochondrial gene expression percentage was calculated for each cell. Cells were filtered based on the criteria nFeature_RNA > 200, nFeature_RNA < 6000, percent.mt < 10, and $\log_{10}$GenesPerUMI > 0.8. Violin plots and scatter plots were generated to visualize nFeature_RNA, percent.mt, and $\log_{10}$GenesPerUMI. The data were then normalized, and the top 2000 highly variable genes were identified. Next, all genes were standardized, followed by principal component analysis (PCA), with the gene loadings and distribution of the top two principal components visualized. The top 30 principal components were selected for downstream analysis, including constructing a neighbor graph and performing cell clustering analysis, with the resolution set to 0.5. UMAP dimensionality reduction was carried out using the 'RunUMAP' function, and the UMAP clustering plot was generated. Batch effect correction was performed using the 'harmony' package, followed by reclustering after UMAP reduction. Harmony was run with default parameters ($\theta = 2$, $\lambda = 1$, $\sigma = 0.1$, block.size = 0.05, max. iter.cluster = 20, max.iter.harmony = 10). Finally, differentially expressed genes for each cell cluster were identified using the 'FindAllMarkers' function, and cell types were annotated based on the identified markers using the CellMarker [16] database.

# Results

## Tryptophan metabolism pathway expressed differentially in carotid plaque

We first preprocessed data from GSE43292. To ensure data normalization, we used the limma method to remove technical variability (Fig 1A-B). Next, the differential expression levels of all genes were analyzed using the limma method. Based on the criteria of |log2FC| > 0.75 and p-value < 0.05, we identified 446 DEGs (Fig 1C, S1 Table). To further investigate the biological functions and potential pathways involved, we performed GO and KEGG pathway enrichment analyses on the identified DEGs. The GO enrichment analysis revealed that these DEGs were significantly enriched in 316 GO functional terms (Fig 1D, S2 Table). Meanwhile, the KEGG analysis identified 53 significantly enriched pathways (Fig 1E, S3 Table), indicating that these DEGs play key roles in various biological signaling pathways.

Both in the GO and KEGG enrichment results, it is demonstrated that there are generally differentially expressed immune and inflammation-related pathways in the carotid plaque samples, such as myeloid leukocyte activation (GO: 0002274, p.adj.value = 6.26E-14), immune response-activating cell surface receptor signaling pathway (GO: 0002429, p.adj.value = 7.83E-14), Complement and coagulation cascades (hsa04610, p.adj.value = 1.50E-06), and B cell receptor signaling pathway (hsa04662, 9.49E-06). This is consistent with the known findings that carotid plaque, although not an immune disease, is closely related to immune response and inflammation and belongs to an immune-mediated pathological process [17]. In addition, it has been also discovered that tryptophan metabolism undergoes alterations in atherosclerosis [18]. We likewise found that the DEGs in carotid plaques were enriched in tryptophan metabolism (hsa00380, p.adj.value = 0.05), aromatic amino acid metabolic process (GO: 0009072, p.adj.value < 0.03), and aromatic amino acid family catabolic process (GO: 0009074, p.adj.value < 0.03), which indicates that the tryptophan metabolic pathway might

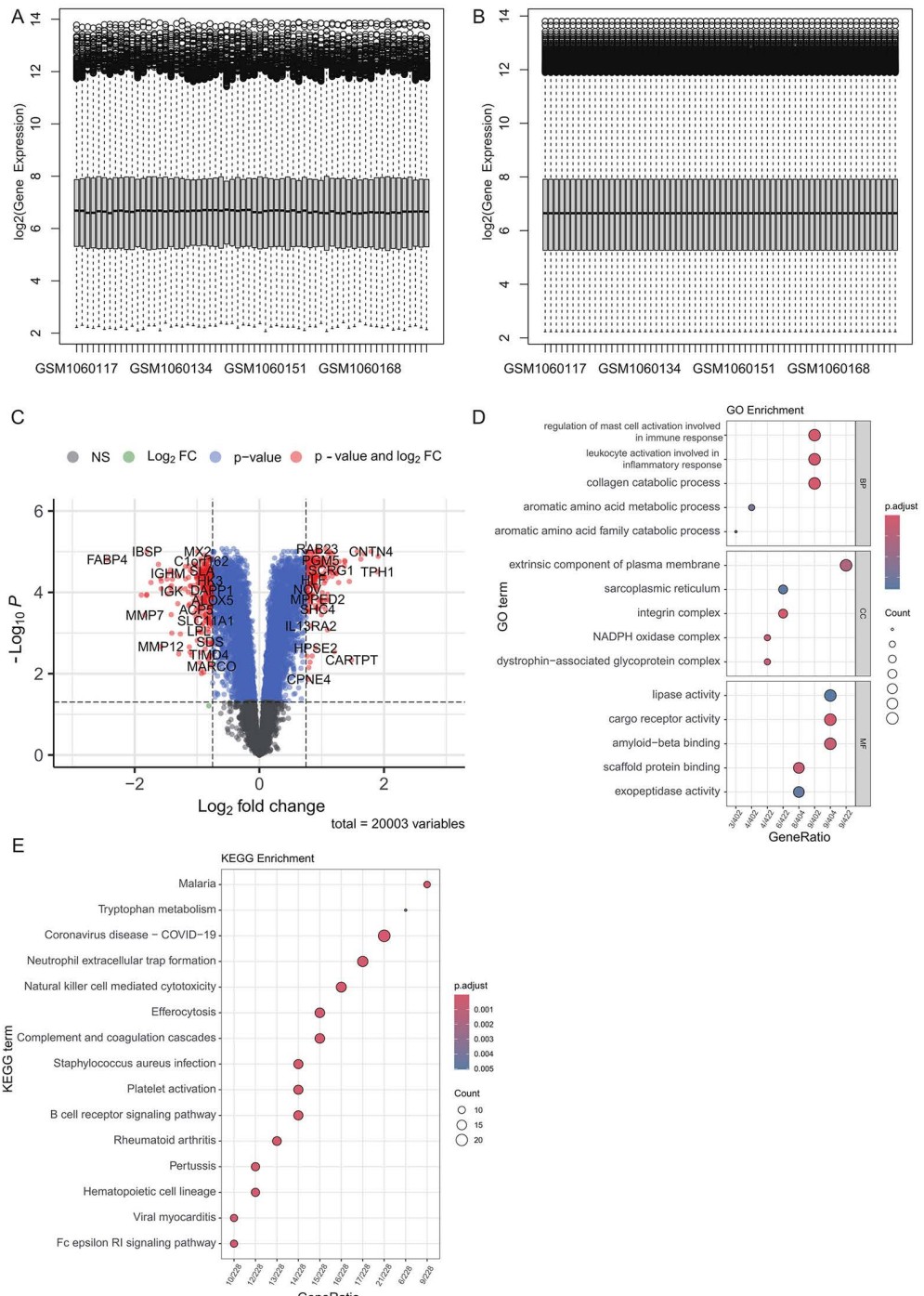

**Fig 1. Overview of differential expression and enrichment analysis between carotid plaque and normal tissue. (A)** and **(B)** show the consistency of samples before and after limma normalization. **(C)** Differentially expressed genes between carotid artery plaques and normal samples, with significance thresholds: |log2FC|>0.75 and p.value<0.05. **(D)** GO enrichment analysis of differentially expressed genes in carotid artery plaques. **(E)** KEGG pathway enrichment of differentially expressed genes in carotid artery plaques.

play a crucial role in carotid plaques. Although the P-value of the enrichment analysis for the tryptophan metabolic pathway did not reach statistical significance (p.adj.value = 0.97), based on the standardized enrichment score (NES), it could be observed that this pathway showed an upregulation trend in plaque tissues (NES = 0.54). This trend suggests that although the sample size is limited and the statistical significance is insufficient, tryptophan-related genes may be in an activated state as a whole in atherosclerotic plaques.

## Tryptophan-related genes as biomarkers of carotid plaque

In this study, we focused on tryptophan metabolism as the primary research direction and collected a set of 50 genes related to tryptophan metabolism based on the literature [19] (S4 Table). By analyzing the cross-relationship between DEGs and genes related to tryptophan metabolism, we prioritized the genes based on three criteria: (1) There was a significant difference in expression between carotid plaque and normal samples (adjusted p value < 0.05, log2FC > 0.75); (2) Included in the gene set related to tryptophan metabolism. Applying these standards, we ultimately identified six key TRDEGs -TPH1, MAOB, TDO2, KMO, KYNU and CYP1B1 - laying the foundation for subsequent analysis (Fig 2A).

The logistic regression model is a statistical model utilized for binary or multi-classification issues, capable of predicting the likelihood of a sample belonging to a certain category by estimating the probability relationship between the independent variables and the classes. We employed six TRDEGs to construct a logistic regression model for predicting the probability of a sample suffering from carotid artery plaques. The GSE43292 dataset was randomly divided into 80% for training and 20% for testing. The training data was used to develop a logistic regression model, with the resulting model coefficients displayed in Table 1. Our model demonstrated strong performance in both the training and test sets (Fig 2B-C). Next, we use ROC and confusion matrix to verify the efficiency of the model. The AUC value of the model is 0.75 (Fig 2D), and the sensitivity and specificity of the confusion matrix are both 0.83. A model constructed using TRDEGs through logistic regression can provide us with an effective way to predict the likelihood of a patient developing carotid plaque and, in turn, the patient's disease risk.

The logistic regression classifier achieved an AUC of 0.82, with a PR-AUC of 0.82, and a Brier score of 0.19, indicating good discrimination and acceptable probability calibration. To further verify that the predictive value of the six-gene panel was not model-dependent, three additional classifiers (LASSO, Random Forest, and SVM) were applied using the same cross-validation strategy. All models demonstrated comparable performance, with AUC values ranging from 0.80 to 0.83, PR-AUC values between 0.71 and 0.82, and Brier scores between 0.18 and 0.19 (Table 2). ROC and PR curves showed similar shapes across all four classifiers (Fig 2E-F), supporting that the six-gene signature provides consistent and reproducible predictive ability for distinguishing carotid plaque from normal tissue, independent of algorithm choice.

## The important role of TRDEGs in carotid plaques

TRDEGs are widely recognized as key genes involved in the mechanisms underlying carotid plaque formation. We found that the expression levels of TPH1, MAOB, TDO2, KMO, KYNU, and CYP1B1 were not consistent in carotid plaques. TDO2, KMO, KYNU and CYP1B1 are upregulated in carotid plaques, while TPH1 and MAOB are downregulated (Fig 3A). This suggests that the pathways and ways in which these TRDEGs participate in regulation are not consistent. KMO [20], KYNU [21], and TDO2 [22] are involved in the development of carotid artery plaques through the tryptophan-kynurenine metabolic pathway; TPH1 competes for tryptophan with the kynurenine pathway by producing serotonin [23]; CYP1B1 contributes to the formation of carotid plaques by participating in the synthesis of cholesterol and other lipids [24]; while MAOB promotes atherosclerosis and carotid plaque formation by damaging vascular endothelial cells and inducing inflammation in the vascular system [25].

Next, we will perform an immune-related analysis on the bulk data of carotid artery plaques. CIBERSORT is a computational tool used to estimate the proportions of different cell types in a complex tissue sample based on gene expression data, particularly useful for characterizing immune cell composition in bulk transcriptomic datasets. The results of

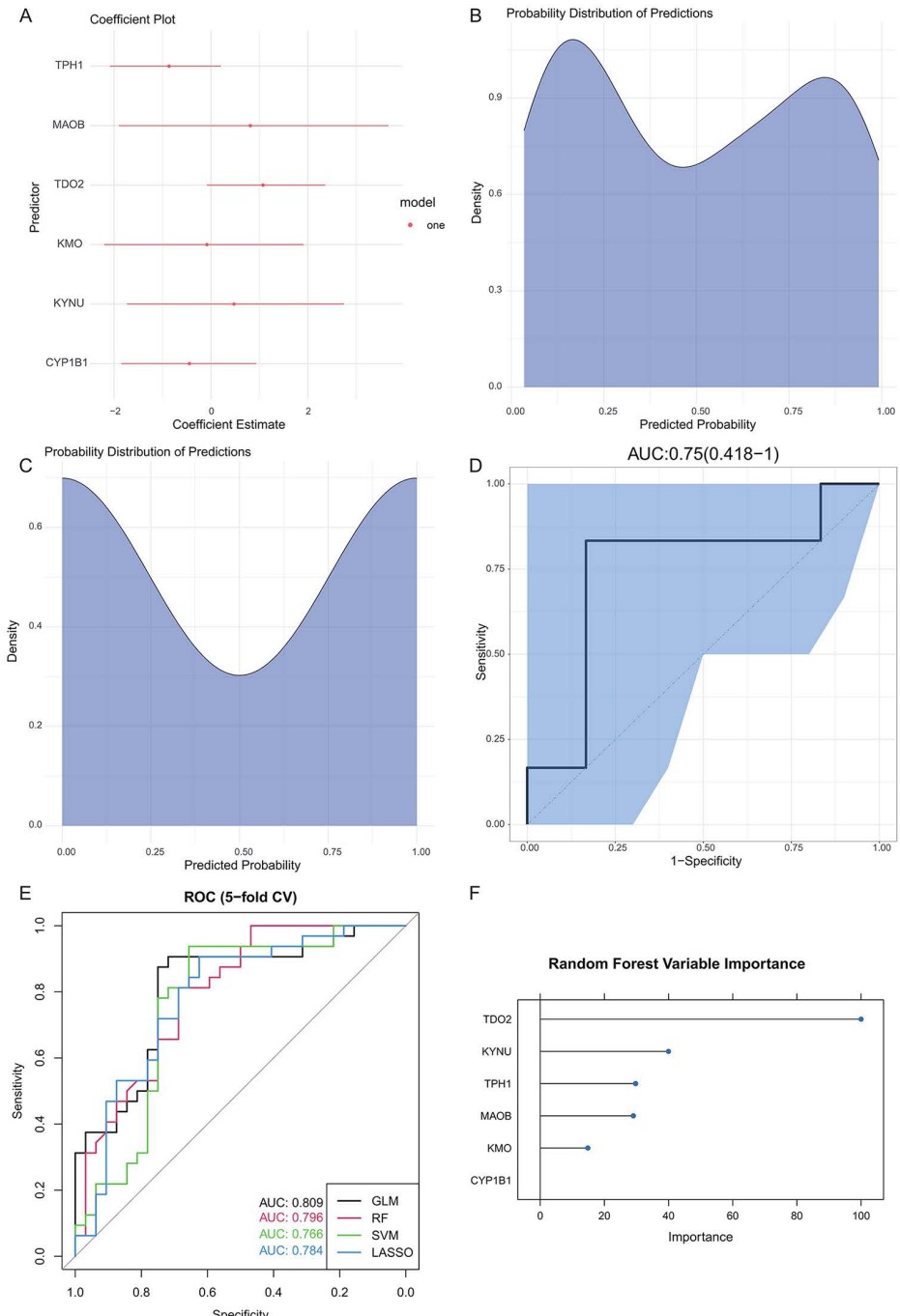

**Fig 2. Construct logistic regression models using TRDEGs. (A)** Coefficient graphs of TRDEGs in logistic regression model. **(B)** Predicted probability distribution of logistic regression model in training set. **(C)** Predicted probability distribution of logistic regression model in test set. **(D)** ROC curve and AUC value of logistic regression model. **(E)** Receiver operating characteristic (ROC) curves of the four classification models (GLM, LASSO, Random Forest, and SVM) constructed using the six TRDEGs. Model performance was evaluated by stratified 5-fold cross-validation, and the area under the ROC curve (AUC) was calculated for each classifier. **(F)** Precision-recall (PR) curves and calibration results of the four TRDEG-based models. PR curves reflect classifier performance in imbalanced settings, while calibration curves evaluate the agreement between predicted and observed probabilities.

**Table 1. The regression coefficients of the key TRDEGs.**

| TRDEGs | Coefficient | LowerCI | UpperCI | P.value | OR |
|--------|-------------|---------|---------|---------|-----|
| TPH1 | 0.42 | 0.12 | 1.23 | 0.13 | 0.42 |
| MAOB | 2.26 | 0.15 | 39.05 | 0.557 | 2.26 |
| TDO2 | 2.93 | 0.92 | 10.62 | 0.0785 | 2.93 |
| KMO | 0.92 | 0.11 | 6.78 | 0.936 | 0.92 |
| KYNU | 1.61 | 0.18 | 15.61 | 0.671 | 1.61 |
| CYP1B1 | 0.64 | 0.16 | 2.55 | 0.519 | 0.64 |

**Table 2. Performance metrics of four TRDEG-based models.**

| Model | AUC | PR_AUC | Brier |
|-------|-----|--------|-------|
| GLM | 0.82 | 0.82 | 0.19 |
| LASSO | 0.80 | 0.73 | 0.19 |
| RandomForest | 0.82 | 0.76 | 0.18 |
| SVM | 0.83 | 0.71 | 0.18 |

CIBERSORT indicate that the proportions of memory B cells, activated CD4$^+$ memory T cells, and M0 macrophages in carotid plaque tissues are significantly increased (Wilcox test, p value < 0.05), while the proportions of CD8$^+$ T cells, activated NK cells, monocytes, and activated dendritic cells are significantly decreased (Wilcox test, p value < 0.05, Fig 3B). The changes in the proportions of these immune cells highlight the immune differences between the normal and carotid plaque samples. Based on the CIBERSORT results, we further assessed the correlations between TRDEG expression levels and the immune cells (Fig 3C). From the heat map, we discovered that the first group of TRDEGs, namely KYNU, KMO, TDO2, and CYP1B1, tended to have a consistent correlation with each type of immune cell; the second group of TRDEGs, namely TPH1 and MAOB, also tended to have a consistent correlation with each type of immune cell, and the correlations of these two genes with the immune cells presented an opposite situation. This phenomenon might be due to the fact that TRDEGs influence the variations in the immune microenvironment of the carotid plaque tissue through a common mechanism, tryptophan metabolism. Meanwhile, these results suggest that TRDEGs can influence immunity through the same mechanisms, but the regulatory mechanisms each TRDEG plays are not the same.

## The construction of the TRDEGs PPI network

The logistic regression model constructed by TRDEGs exhibits excellent performance in predicting carotid artery plaques and we also found that TRDEGs can influence the immune response of carotid artery plaques through the tryptophan metabolic pathway, however, we aim to further explore the interactions between these TRDEGs to gain a deeper understanding of the role of tryptophan metabolism in the development of carotid artery plaques. Therefore, we imported the six TRDEGs into the STRING database to generate a PPI network of TRDEGs (Fig 4A). After obtaining the PPI network for the TRDEGs, we calculated the degree centrality, betweenness centrality, and closeness centrality of each gene using the exported interaction data (Table 3). These metrics were used to assess the importance of each gene within the network. In the PPI network, KMO, TDO2, and TPH1 occupy more central and influential positions compared to other nodes. The elevated centrality metrics indicate that these genes play a crucial role in maintaining the structure and function of the network.

Furthermore, the STRING database demonstrates the interaction methods between proteins through the edges in the network. The light blue lines indicate that there are "Known Interactions" between proteins from "from curated databases". It can be observed from Fig 4A that the known interactions exist among four pairs of proteins: KYNU – KMO, KYNU

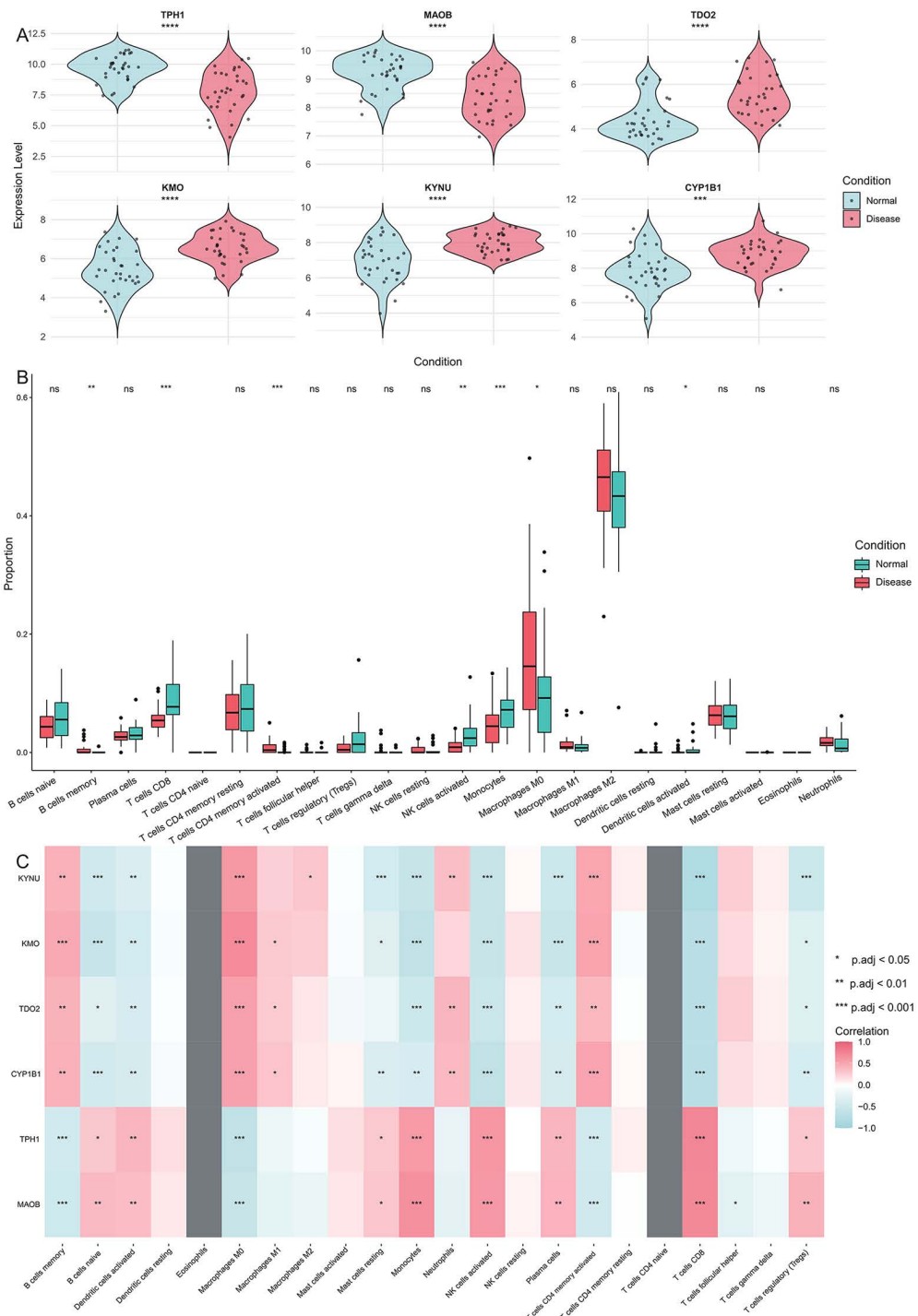

**Fig 3. Expression patterns of TRDEGs and their relationship with immunity. (A)** The expression difference of six TRDEGs in carotid plaque and normal tissue, and the differential expression of all genes met significance thresholds: |log2FC|>0.75 and p.value<0.05. Statistical significance between groups was determined using the Wilcoxon rank-sum test. **(B)** Differences in immune cell infiltration proportions between carotid plaque and normal tissue were evaluated using the Wilcoxon rank-sum test. * P<0.05, ** P<0.01, *** P<0.001, ns non-significant. **(C)** Heatmap of correlations between TRDEGs' expression and immune cell infiltration proportions. Correlation analysis was performed using the Spearman rank correlation method. * p.adj<0.05, ** p.adj<0.01, and *** p.adj<0.001.

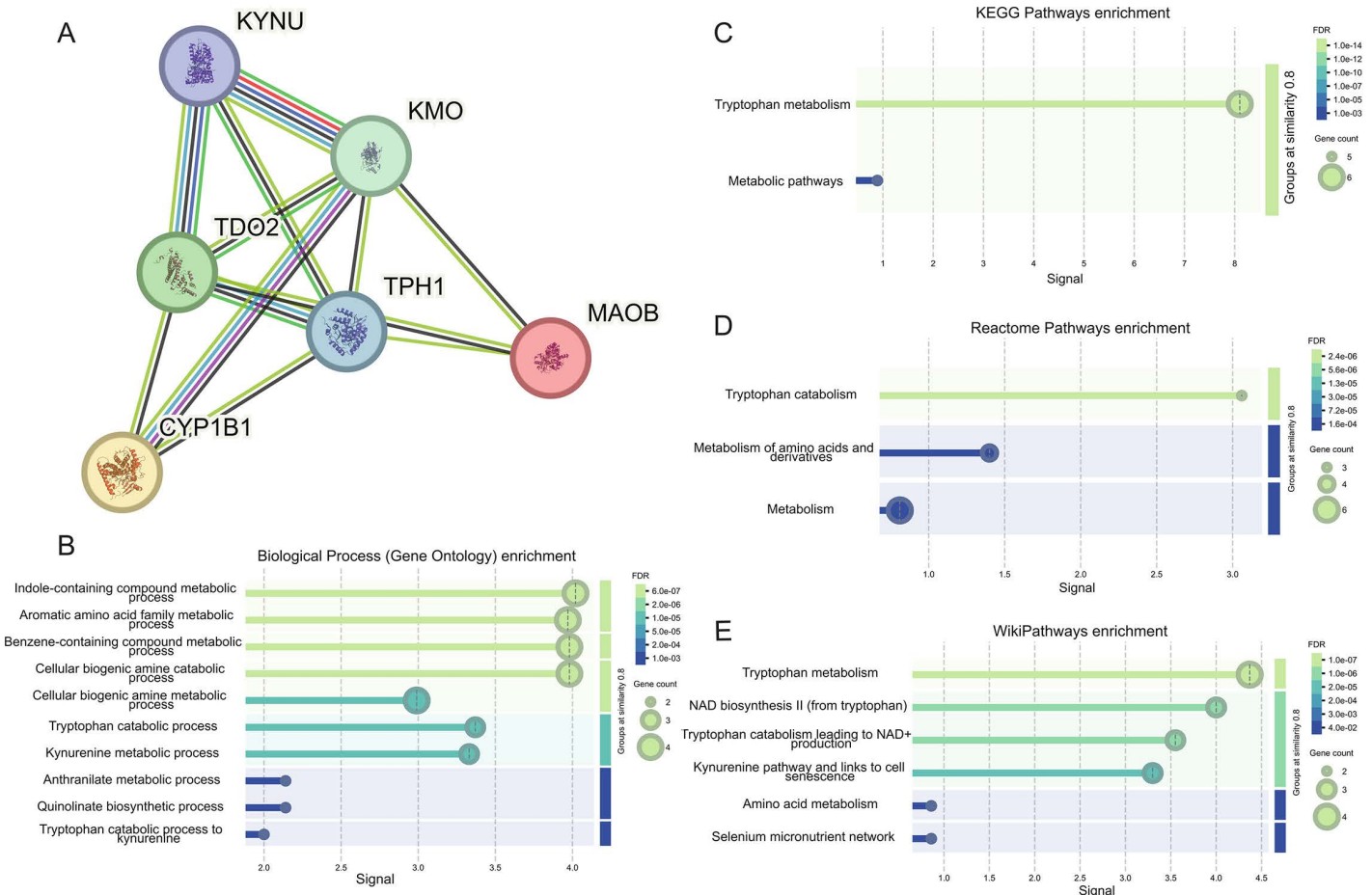

**Fig 4. Construction of PPI network and its enriched functions and pathways. (A)** A PPI network built using a STRING database with TRDEGs as input. Nodes represent proteins encoded by these TRDEGs, while edges represent predicted or known interactions, including direct (physical) and indirect (functional) associations. The thickness of the edge indicates confidence in the interaction based on the available evidence. **(B)** Result of Biological Process (Gene Ontology) enrichment of this network analyzed by STRING. **(C)** Result of KEGG Pathways enrichment results for this network analyzed by STRING. **(D)** Results of Reactome Pathways enrichment of this network analyzed by STRING. **(E)** Result of WikiPathways enrichment results for this network analyzed by STRING.

**Table 3. The PPI information of the key TRDEGs.**

| TRDEGs | Degree | Betweenness | Closeness |
|---|---|---|---|
| KMO | 10 | 1 | 1 |
| TDO2 | 10 | 1 | 1 |
| TPH1 | 10 | 1 | 1 |
| CYP1B1 | 6 | 0 | 0.71 |
| KYNU | 6 | 0 | 0.71 |
| MAOB | 6 | 0 | 0.71 |

- TDO2, KMO - CYP1B1, and TDO2 - TPH1. In the kynurenine metabolic pathway, KMO, as the upstream of KYNU, directly participates in metabolic regulation [26]. Meanwhile, TDO2, as the first step in tryptophan metabolism, regulates the expressions of KMO and KYNU [27]. There is no direct regulatory relationship between KMO and CYP1B1. However, in addition to their involvement in the tryptophan catabolic pathway [28], they are both AHR target genes, and their expressions can be upregulated through anakinra and AHR signaling [29]. There is also no direct regulatory relationship between TDO2 and TPH1. TDO2 is mainly responsible for catalyzing tryptophan to enter the kynurenine pathway and converting it into kynurenine; while TPH1 catalyzes the conversion of tryptophan into serotonin. However, an indirect regulatory relationship may exist between TDO2 and TPH1 through the competition for the substrate, tryptophan [30].

Finally, we exported the functional and pathway enrichment of the PPI network from the STRING database (Fig 4B-E). The enrichment results from four different databases indicate that the network is primarily enriched in functions and pathways related to tryptophan metabolism and kynurenine metabolism, which is consistent with our previous findings.

## Investigating the role of TRDEGs in carotid plaques through scRNA analysis

In order to find out exactly in which cells and how TRDEGs affect the occurrence of carotid plaque by regulating different pathways, we used the data in GSE159677 to construct a scRNA landscape of carotid plaque samples. First, we preprocessed the single-cell data and removed the batch effect (Fig 5A). Next, the single-cell data were clustered, and cell types were annotated using CellMarker database for differentially expressed genes in each cluster. We ended up with nine cell types that were dominated by immune cells, including T cell, endothelial cell, smooth muscle cell, dendritic cell, macrophage, fibroblast, B cell, natural killer cell and granulocyte-monocyte progenitor (Fig 5B-C).

We calculated the expression of TRDEGs in each cell type, as shown in Fig 5D. The expressions of KMO, TDO2, CYP1B1, KYNU and MAOB in single-cell data were basically consistent with those in bulk data. Specifically, TPH1 expression was down-regulated in bulk data, but up-regulated in carotid plaque fibroblast and smooth muscle cell. We looked at the difference in the number of the two types of cells between carotid plaque and normal samples. We next assessed cell-type composition in carotid plaques using scRNA-seq. Although no cell type reached statistical significance after FDR correction (Fig 5E; S5 Table), endothelial cells, fibroblasts, and smooth muscle cells showed decreasing proportions in plaques, whereas macrophages and dendritic cells exhibited an increasing trend. These compositional shifts provide a plausible explanation for TRDEG expression patterns. Five genes (KMO, TDO2, CYP1B1, KYNU, and MAOB) were consistent between bulk and single-cell data, while TPH1 was elevated in fibroblasts and smooth muscle cells but downregulated in bulk tissue, likely due to the reduced abundance of these cell types in plaques. Together, these findings indicate that TRDEG heterogeneity is shaped by both transcriptional changes and cell-type proportion shifts within the plaque microenvironment.

## Discussion

KMO, KYNU and TDO2 are the key regulatory genes of tryptophan-kynurenine metabolism. KMO is a gene that encodes an enzyme involved in the kynurenine pathway, which is the main metabolic pathway for tryptophan degradation. The 3-hydroxykynurenine produced by KMO can activate innate immune signals, thereby exacerbating systemic inflammation. The upregulation of KMO expression suggests that the kynuurine pathway in carotid artery plaque tissue may be upregulated, inducing the occurrence of carotid artery plaques. KYNU is a hydrolase that promotes the decomposition of tryptophan in eukaryotes through the kynurenine pathway. TDO2 catalyzes the initial and rate-limiting steps of tryptophan degradation, converting tryptophan into N-formylkynurenine, which is then metabolized into kynurenine and other downstream metabolites. Through the PPI network, we determined that the regulatory relationship among KYNU, KMO and TDO2 also exists in carotid artery plaques. Furthermore, we believe that the activity of the tryptophan – kynurenine metabolic pathway significantly increases in carotid plaques and may induce immunosuppression by inhibiting T cell activity. The excessive activation of the kynourine pathway may lead to the accumulation of metabolites, causing oxidative

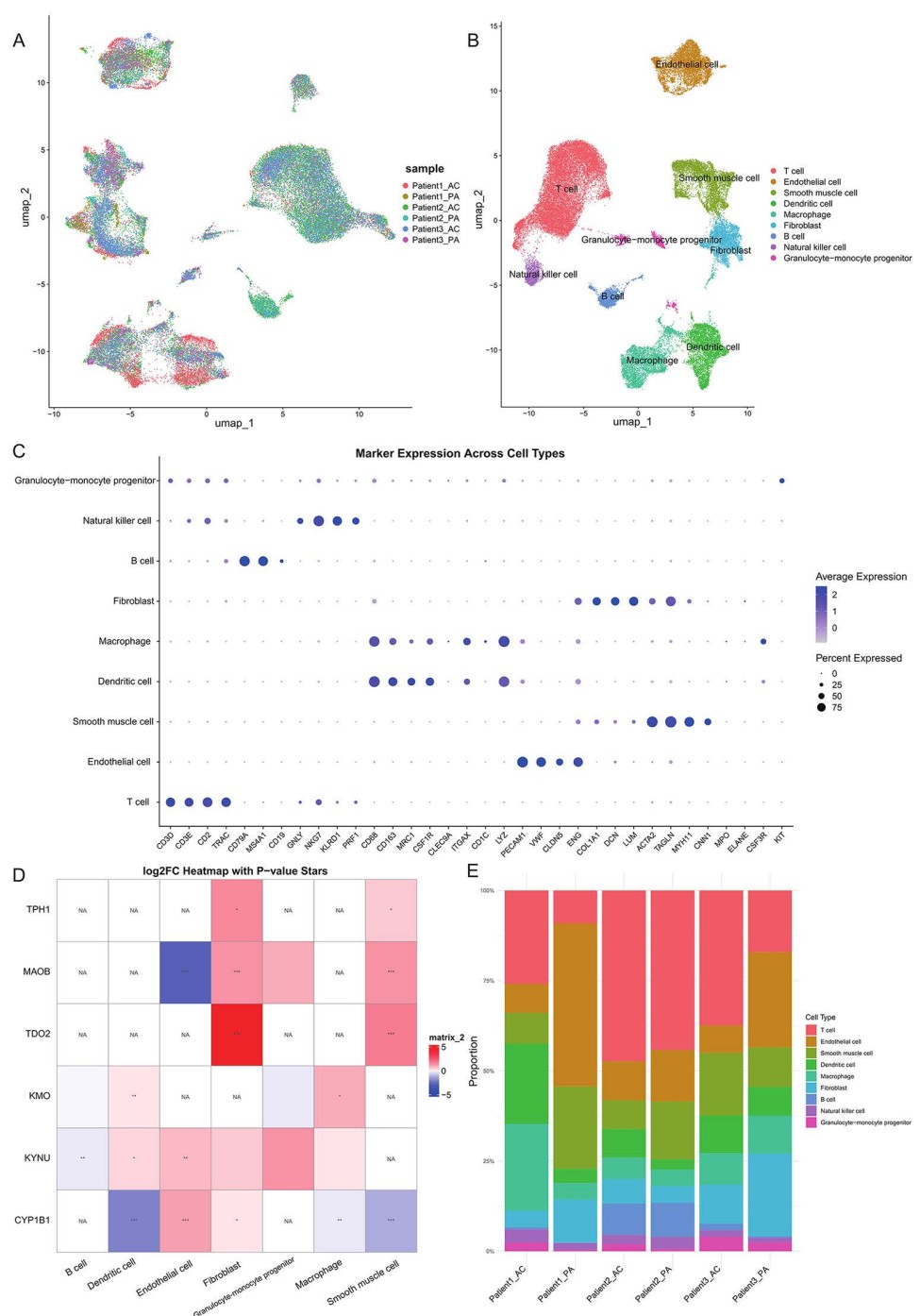

**Fig 5. Construction of carotid plaque single cell transcriptome data landscape. (A)** Umap distribution of single-cell data after pretreatment and batch removal, with the figure colored according to cell source. **(B)** Umap distribution map after annotated cell types from single cell data, colored according to annotated cell types. **(C)** Expression patterns of canonical marker genes across annotated cell types in the single-cell dataset. **(D)** Heatmap of TRDEGs expression changes in each cell type. No TRDEGs were differentially expressed in T cells and Natural Killer cells, so these cell types were removed from the heatmap. * P < 0.05, ** P < 0.01, *** P < 0.001, ns non-significant. **(E)** Histogram of cell proportions showing the differences in each cell type in different samples.

stress and endothelial cell damage, thereby increasing the risk of atherosclerosis progression [31]. TPH1 catalyzes the conversion of tryptophan to serotonin through 5-HTP. Serotonin produced by TPH1 in platelets plays an important role in vasoconstriction and coagulation. Under inflammatory conditions, elevated TDO2 activity introduces more tryptophan into the kynurenine pathway, thereby reducing the supply of tryptophan required for serotonin synthesis, which indirectly inhibits TPH1 activity and indirectly leads to a decrease in TPH1 expression [32]. Regarding the expression of TPH1, some differences have emerged between bulk RNA-seq and scRNA-seq. In scRNA-seq, the expression of TPH1 is upregulated in fibroblasts and smooth muscle cells, but the proportion of these two types of cells in carotid artery plaques is lower than that in normal tissues, which may lead to the down-regulation of TPH1 expression in bulk RNA-seq. The specific expression pattern of TPH1 in carotid artery plaques is also worthy of attention. Regarding the interaction between TPH1 and TDO2 in the PPI network, we propose that TDO2 mainly catalyzes tryptophan to enter the kynuurine pathway, converting it into kynuurine, while TPH1 converts tryptophan into serotonin. Therefore, TDO2 and TPH1 may have an indirect regulatory relationship by competing for tryptophan. Meanwhile, MAOB is a 5-HT degrading enzyme and belongs to the downstream regulatory factor of the tryptophan – serotonin branch [33]. However, the reduction of the tryptophan – serotonin metabolic pathway may have affected the expression of MAOB, but the specific regulatory direction remains to be further determined. Finally, kynurenine can also act as an endogenous AhR agonist [34]. CYP1B1 (Cytochrome P450 1B1) is a downstream target gene of AHR and is involved in the metabolism of cholesterol and other lipids [35], and these lipids are key components in the occurrence of atherosclerosis (Fig 6).

In this study, we identified multiple TRDEGs (such as TDO2, KMO and KYNU) that jointly point to the abnormal activation of the tryptophan kynuurine pathway, which is the main direction of tryptophan catabolic metabolism and has been continuously confirmed in recent years to be closely related to the occurrence and development of atherosclerosis. In human atherosclerotic plaques, IDO1 is significantly upregulated in macrophage-rich regions and is associated with

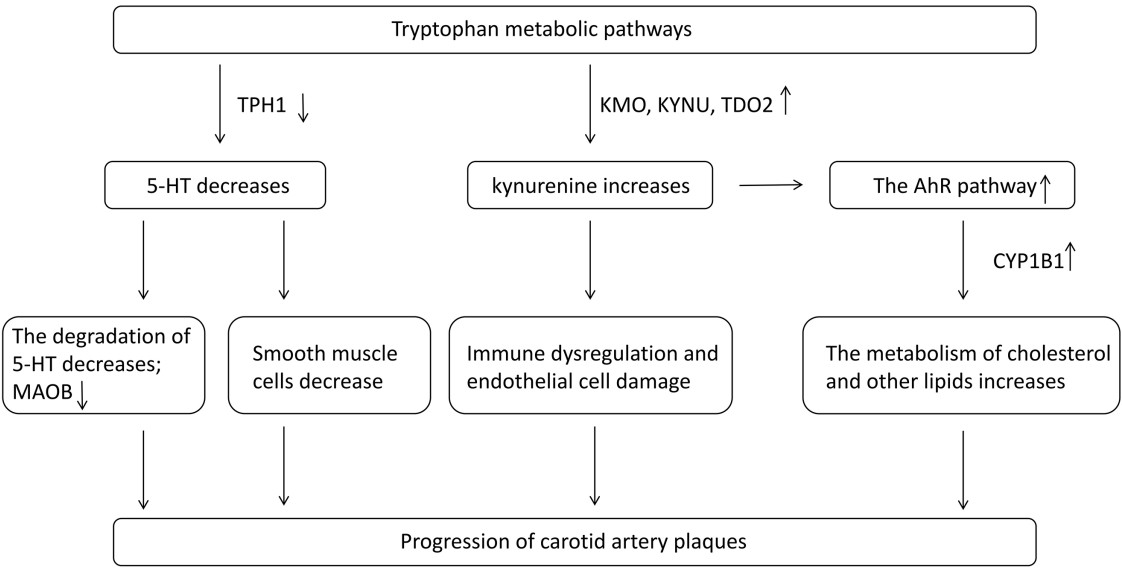

**Fig 6. The proposed mechanisms linking TRDEGs to immune dysregulation and plaque progression in carotid atherosclerosis.** This schematic illustrates how six key TRDEGs coordinate metabolic and immune alterations in carotid atherosclerotic plaques. Up-regulated kynurenine pathway enzymes (TDO2, KMO and KYNU) enhance kynurenine production, and activate the AhR signaling pathway. Meanwhile, serotonin synthesis is reduced, accompanied by decreased smooth muscle cells, while MAOB-mediated 5-HT degradation promotes oxidative stress and endothelial dysfunction. In parallel, CYP1B1 participates in AhR-dependent lipid metabolic reprogramming, further aggravating plaque formation. Together, these TRDEG-driven metabolic changes disrupt vascular homeostasis and accelerate the progression of carotid artery plaques.

plaque instability by enhancing kyurine metabolism and promoting the expression of local tissue factors [36]. In line with human studies, animal experiments have further confirmed that when the IDO1-kyrie acid axis is disrupted, it accelerates vascular calcification and plaque formation. Exogenous supplementation of kynurenine can reverse the above adverse changes through immune metabolic signal regulation [31]. These pieces of evidence collectively indicate that the disorder of tryptophan metabolism is a key regulatory link closely coupled with the remodeling of the immune microenvironment of carotid plaques.

Atherosclerotic plaques typically present a chronic inflammatory microenvironment characterized by infiltration of macrophages, T cells and dendritic cells. Metabolic reprogramming of the kynuurine pathway can be achieved by consuming local tryptophan and generating various immunomodulatory metabolites (such as Kyn and 3-HK). It triggers T-cell exhaustion and macrophage phenotypic polarization, thereby driving immunosuppression and the persistence of chronic inflammation. Therefore, the abnormal expression of TRDEGs may be a key molecular hub connecting metabolic imbalance and immune disorder, providing a new explanatory framework for us to understand the "metabolism-immune crosinterference" in atherosclerosis.

Despite the initial insights offered by this study regarding tryptophan-related gene changes in carotid atherosclerotic plaques, several limitations should be acknowledged. First, our analysis relied on a single bulk RNA-seq dataset. The absence of validation in additional independent human cohorts or animal models limits the robustness and generalizability of the identified differential gene signatures. Second, although we observed biological trends relevant to the study's hypothesis, the enrichment of the tryptophan metabolism pathway did not reach statistical significance, which is likely attributable to the limited sample size of the current dataset. Third, the immune infiltration estimates and single-cell type annotations were derived exclusively through computational algorithms. While these methods provide a systems-level perspective, the lack of experimental corroboration may introduce potential bioinformatic biases. Finally, this study remains descriptive in nature, as no functional assays were performed to mechanistically confirm the specific roles of the identified genes in the progression of atherosclerosis. Consequently, these findings should be interpreted with caution, and future studies incorporating larger sample sizes and experimental validation are warranted to verify these preliminary results.

In summary, we identified six TRDEGs from the logistic regression model that participate in the formation and progression of carotid atherosclerotic plaques through different mechanisms related to tryptophan metabolism, lipid biosynthesis, and inflammatory responses. These TRDEGs play a crucial role in regulating key pathways involved in plaque formation and progression, not only serving as predictive markers for assessing the risk of carotid atherosclerotic plaque formation but also having the potential to be therapeutic targets. Targeting these genes may provide new strategies for the treatment and management of carotid atherosclerotic plaques and improving clinical outcomes.

## Supporting information

**S1 Table. Differentially expressed genes.**
(XLSX)

**S2 Table. The result of GO functional enrichment.**
(XLSX)

**S3 Table. The result of KEGG pathway enrichment.**
(XLSX)

**S4 Table. Tryptophan-related signature genes.**
(XLSX)

**S5 Table. Cell-type composition comparison between carotid plaque (CA) and proximal artery (PA) samples.**
(XLSX)

## Author contributions

**Conceptualization:** Qiang Zhang, Runze Jiang.

**Data curation:** Qiang Zhang, Xiaodong Jia.

**Formal analysis:** Xiaodong Jia, Lin Bai, Xianchao Guo.

**Investigation:** Zhen Wang, Jianmei Wei.

**Methodology:** Qiang Zhang, Xiaodong Jia, Zhen Wang, Lin Bai, Xianchao Guo, Runze Jiang.

**Software:** Lin Bai, Xizi Wang, Zhaona Song, Xianchao Guo.

**Validation:** Jianmei Wei, Xizi Wang, Zhaona Song.

**Visualization:** Lin Bai, Jianmei Wei, Xizi Wang, Zhaona Song.

**Writing – original draft:** Xiaodong Jia, Lin Bai.

**Writing – review & editing:** Qiang Zhang, Zhen Wang, Runze Jiang.

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
