## [Decision Letter · Decision Letter 0]

17 Sep 2025

Dear Dr. zhang,

Thank you for submitting your manuscript to PLOS ONE. After careful consideration, we feel that it has merit but does not fully meet PLOS ONE’s publication criteria as it currently stands. Therefore, we invite you to submit a revised version of the manuscript that addresses the points raised during the review process.

We look forward to receiving your revised manuscript.

Kind regards,

Adeyemi Stephen Stephen Oluyomi

Academic Editor

PLOS ONE

Journal Requirements:

Reviewers' comments:

Reviewer's Responses to Questions

**Comments to the Author**

1. Is the manuscript technically sound, and do the data support the conclusions?

Reviewer #1: Yes

Reviewer #2: Yes

Reviewer #3: Yes

2. Has the statistical analysis been performed appropriately and rigorously?

Reviewer #1: Yes

Reviewer #2: Yes

Reviewer #3: No

3. Have the authors made all data underlying the findings in their manuscript fully available?

Reviewer #1: Yes

Reviewer #2: Yes

Reviewer #3: Yes

4. Is the manuscript presented in an intelligible fashion and written in standard English?

Reviewer #1: Yes

Reviewer #2: Yes

Reviewer #3: Yes

Reviewer #1: The manuscript by Jia et al. describes the in silico identification of tryptophan metabolism–related genes involved in carotid artery plaque progression using bulk and single-cell RNA sequencing data from carotid plaque patients. Among 446 differentially expressed genes, the authors focused on six key tryptophan-related genes including TPH1, MAOB, TDO2, KMO, KYNU, and CYP1B1. They constructed a logistic regression model with an AUC of 0.75, which predicted the risk of carotid plaque formation. Analyses of bulk and single-cell data further revealed differential expression patterns and potential mechanisms of action of these genes in carotid plaque, suggesting roles in tryptophan metabolism, lipid biosynthesis, and inflammatory responses.

This study has the potential to improve the understanding of how tryptophan metabolism contributes to carotid plaque progression, identify new biomarkers for risk assessment, and highlight possible molecular targets for therapeutic intervention. However, I have a few suggestions to further improve the manuscript:

The full names of Kyn and 5-HT in line 55, as well as TDO and IDO in line 57, are missing.

The second paragraph (introducing the three tryptophan metabolism pathways) and the third paragraph (describing their relationships to carotid artery plaques) could be combined to provide a more concise rationale.

The language is generally clear, but there are several grammar mistakes, and a few long sentences should be shortened to improve readability.

TPH1, MAOB, TDO2, KMO, KYNU, and CYP1B1 were identified as tryptophan metabolism–related genes involved in carotid plaque progression, and their potential mechanisms are well discussed. Including a schematic figure at the end to illustrate the proposed mechanisms would further strengthen the manuscript and help readers grasp the main findings.

Reviewer #2: Metabolism-Related Genes in Carotid Artery Plaques" analyzes tryptophan metabolism-related differential genes in human atherosclerotic plaques. It explores the correlation between these differential tryptophan metabolism genes and atherosclerotic lesions, while analyzing their expression patterns across different cell types using single-cell RNA-seq data. However, I have some suggestions on the manuscript before further consideration.

1.The study mainly analyzed the correlation between tryptophan metabolism related genes and atherosclerotic lesions.

The study's primary objective, as indicated by the title, was to identify tryptophan metabolism-related differential genes in atherosclerotic plaques. However, the identification of differential genes relied solely on a single RNAseq dataset without validation. While enrichment analysis was conducted for these genes, the p-value for tryptophan metabolism pathway enrichment reached 0.05. Moreover, the combined analysis of upregulated and downregulated genes failed to clarify whether tryptophan metabolism pathways were downregulated or upregulated during disease progression. Although logistic regression models attempted to demonstrate correlations between tryptophan metabolism-related differential genes and atherosclerosis, the small sample size resulted in an AUC value of merely 0.75. Correlation analyses aimed to establish links between these differential genes and plaque immune cell infiltration phenotypes, but neither immune infiltration phenotypes were validated in single-cell RNAseq data nor did they undergo multiple hypothesis correction. Finally, six amino acid metabolism-related genes showed differential expression across cell types in the single-cell RNAseq dataset. However, no cell type clustering markers were provided, and the absence of validation using known marker gene expression profiles (e.g., CD68 for macrophages) may have introduced bias in cell clustering.

2.The data are insufficient to support the conclusion:

Regarding the identification of differentially expressed genes, the study utilized only a single RANseq dataset, lacking duplicate validation. To ensure reliability, at least two independent human cohort RNAseq datasets should be added as validation sets for repeated verification. Alternatively, differential genes could be validated using either a single human cohort RNAseq dataset or multiple atherosclerotic mouse model RNAseq datasets. Furthermore, RNAseq results can be cross-validated through qPCR, Western blotting, or immunohistochemistry in human or mouse samples. Whether tryptophan metabolic pathways are upregulated or downregulated in disease states warrants investigation. We recommend conducting GSEA enrichment analysis on tryptophan metabolic pathways to confirm this hypothesis.

It is suggested to use in vitro cell models (such as vascular endothelial cells) to verify the effects of TRDEGs on inflammation and lipid metabolism.

Why not validate the aforementioned immune cell infiltration with single-cell RANseq data? Single-cell RNAseq requires the provision of markers to distinguish each cell type, and it needs to be validated using recognized cell-type markers.

It is recommended to provide analytical code to ensure the reproducibility of the results.

3.Defects of data analysis method:

Box plots are required to visualize data distribution patterns, which can be created by adding data points or plotting violin-shaped histograms. The text fails to specify the exact methodology for scRNA batch correction (only mentioning the Harmony package), and should provide detailed parameter settings. Additionally, Figures 3A and 3B lack proper reference to hypothesis testing methodologies.

Reviewer #3: In this manuscript, Jia et al. identified six key Tryptophan-related TRDEGs (TPH1, MAOB, TDO2, KMO, KYNU, and CYP1B1) that hold paramount significance in carotid artery plaques based on total RNA-seq and scRNA-seq. The authors claim that these six genes can serve as potential biomarkers to assess plaque risk.

Although this manuscript is well-written, there are a few issues that the authors may try to clarify, as suggested below:

1. The analysis of this work is based on a simple study (RNA-seq, GSE43292; scRNA-seq, GSE159677) with an imbalance of the patient numbers between these two datasets. Do the authors consider any artifacts of the analyses caused by inter-patient variants? If so, were any steps of the quality control at the experimental and/or analytical steps included? Please comment on it.

2. Line 185: The authors selected 6 TRDEGs from a set of 50 genes without clarifying the selection criteria. Please provide more details about the strategy that indicated 6 TRDEGs were picked up.

3. Line 196: The AUC value of 0.75 is however not optimal. This observation probably reflects the possibility of insufficiency of selected TRDEGs. Aligning with the previous concern, a detailed explanation of the selection criteria is needed. In addition, it is recommended that the authors construct an additional classifier to verify the findings computed from the model constructed by logistic regression.

4. Lines 219 - 221: The description related to Figure 3B escaped me. If I interpreted the plot correctly, the proportion of M0 macrophages showed an increase in carotid plaques, whereas a decrease in the proportions of CD8+ T cells, activated NK cells, monocytes, and activated DCs was detected.

5. Line 230: It is not clear to me about the immune microenvironment mentioned here. Please elaborate more about this immune microenvironment associated with Tryptophan-related pathways triggered by TRDEGs in the Discussion.

6. Line 251: It is not clear to me why, for example, KYNU possesses fewer degrees than other nodes (i.e., KMO, TDO2, and TPH1) in the network; however, it is shown in the network that a high edge connectivity of the node KYNU is demonstrated. Furthermore, it is not clear how “closeness” is calculated here. In principle, the edge connectivity (or correlation coefficients) has to be calculated based on statistical significance (e.g., Pearson correlation). It is therefore not understandable to me why the value can be 1. Nevertheless, please provide more details on how parameters related to the graph network are computed.

7. Line 253: Figure 3A seems to be inconsistent with the text mentioned here. How bar graphs can display the interactions across proteins.

8. Line 284: In Figure 5D, I did observe a noticeable variation across patients marked in the same groups (AC versus PA), resonating with my comment #1. Since variations exist, a proper statistical test is required here.

9. Line 289: Once again, I cannot link Figure 4E to the text here. Figure 4E describes the pathway enrichment, instead of the proportion of cells.

**Do you want your identity to be public for this peer review?** For information about this choice, including consent withdrawal, please see our Privacy Policy

Reviewer #1: No

Reviewer #2: No

Reviewer #3: **Yes:** Heng-Chang Chen

---

## [Author Response · Author response to Decision Letter 1]

27 Oct 2025

Reviewer #1

1.The full names of Kyn and 5-HT in line 55, as well as TDO and IDO in line 57, are missing.

Thank you for your suggestion, we added the full names of Kyn, 5-HT, TDO and IDO in lines 52 to 57 in the revised manuscript.

In lines 52 to 57:

Its levels depend both on dietary intake and the activity of three main metabolic pathways: the kynurenine (Kyn), 5-hydroxytryptamine (5-HT), and indole pathways[5] .The kynurenine pathway, responsible for over 95% of tryptophan degradation, is regulated by the rate-limiting enzymes TDO (Tryptophan 2,3-dioxygenase), IDO1 (Indoleamine 2,3-dioxygenase 1), and IDO2 (Indoleamine 2,3-dioxygenase 2)[6].

2.The second paragraph (introducing the three tryptophan metabolism pathways) and the third paragraph (describing their relationships to carotid artery plaques) could be combined to provide a more concise rationale.

Thank you for your suggestion, we combined the second and third paragraphs of the Introduction to provide a more concise rationale.

In lines 50 to 71:

Tryptophan is an essential amino acid that the human body cannot synthesize and must be obtained through diet. Beyond serving as a building block for proteins, tryptophan participates in neurotransmitter synthesis, immune regulation, and the modulation of sleep and mood. Its levels depend both on dietary intake and the activity of three main metabolic pathways: the kynurenine (Kyn), 5-hydroxytryptamine (5-HT), and indole pathways[5] .The kynurenine pathway, responsible for over 95% of tryptophan degradation, is regulated by the rate-limiting enzymes TDO (Tryptophan 2,3-dioxygenase), IDO1 (Indoleamine 2,3-dioxygenase 1), and IDO2 (Indoleamine 2,3-dioxygenase 2)[6]. This pathway generates kynurenine and related metabolites that modulate inflammation, immune responses, and excitatory neural signaling, and have been linked to various diseases. Alterations in kynurenine pathway activity are associated with carotid plaque formation and instability: Munn et al. reported that changes in key enzyme activity correlate with plaque instability[7], while Shen et al. demonstrated that IDO1 influences atherosclerosis via local inflammation and immune regulation[8]. Kynurenine metabolites may also affect endothelial function, further contributing to plaque development. The 5-HT pathway converts tryptophan into 5-hydroxytryptophan (5-HTP) and subsequently into 5-hydroxytryptamine (serotonin)[9]. While 5-HT functions primarily as a neurotransmitter, it also impacts vascular endothelium and platelets, potentially promoting plaque formation through regulation of vascular tone, smooth muscle proliferation, and platelet activation. The precise contribution of serotonin to carotid atherosclerosis remains under investigation. Finally, the indole pathway, mediated by gut microbiota, produces indole and its derivatives, which help maintain intestinal homeostasis, modulate systemic inflammation, and regulate immune responses[10]. Given that atherosclerosis is part of the systemic inflammatory response, indole metabolites may influence plaque formation and stability by reducing inflammation and protecting vascular endothelium[11,12].

3.The language is generally clear, but there are several grammar mistakes, and a few long sentences should be shortened to improve readability.

Thank you for your suggestion, we revised the manuscript to correct grammatical errors and have shortened overly long sentences to improve clarity and readability.

4.TPH1, MAOB, TDO2, KMO, KYNU, and CYP1B1 were identified as tryptophan metabolism–related genes involved in carotid plaque progression, and their potential mechanisms are well discussed. Including a schematic figure at the end to illustrate the proposed mechanisms would further strengthen the manuscript and help readers grasp the main findings.

Thank you for your suggestion, at the end of the article, we added a schematic figure to illustrate the proposed mechanisms, and in the discussion, we optimized our presentation of the mechanisms based on the schematic figure.

In lines 354-360:

Meanwhile, MAOB is a 5-HT degrading enzyme and belongs to the downstream regulatory factor of the tryptophan - serotonin branch[33]. However, the reduction of the tryptophan - serotonin metabolic pathway may have affected the expression of MAOB, but the specific regulatory direction remains to be further determined. Finally, kynurenine can also act as an endogenous AhR agonist[34]. CYP1B1 (Cytochrome P450 1B1) is a downstream target gene of AHR and is involved in the metabolism of cholesterol and other lipids[35], and these lipids are key components in the occurrence of atherosclerosis (Fig.6).

In lines 392-400:

Fig 6. The proposed mechanisms linking TRDEGs to immune dysregulation and plaque progression in carotid atherosclerosis. This schematic illustrates how six key TRDEGs coordinate metabolic and immune alterations in carotid atherosclerotic plaques. Up-regulated kynurenine pathway enzymes (TDO2, KMO and KYNU) enhance kynurenine production, and activate the AhR signaling pathway. Meanwhile, serotonin synthesis is reduced, accompanied by decreased smooth muscle cells, while MAOB-mediated 5-HT degradation promotes oxidative stress and endothelial dysfunction. In parallel, CYP1B1 participates in AhR-dependent lipid metabolic reprogramming, further aggravating plaque formation. Together, these TRDEG-driven metabolic changes disrupt vascular homeostasis and accelerate the progression of carotid artery plaques.

Reviewer #2

1.The study mainly analyzed the correlation between tryptophan metabolism related genes and atherosclerotic lesions.

The study's primary objective, as indicated by the title, was to identify tryptophan metabolism-related differential genes in atherosclerotic plaques. However, the identification of differential genes relied solely on a single RNAseq dataset without validation. While enrichment analysis was conducted for these genes, the p-value for tryptophan metabolism pathway enrichment reached 0.05. Moreover, the combined analysis of upregulated and downregulated genes failed to clarify whether tryptophan metabolism pathways were downregulated or upregulated during disease progression. Although logistic regression models attempted to demonstrate correlations between tryptophan metabolism-related differential genes and atherosclerosis, the small sample size resulted in an AUC value of merely 0.75. Correlation analyses aimed to establish links between these differential genes and plaque immune cell infiltration phenotypes, but neither immune infiltration phenotypes were validated in single-cell RNAseq data nor did they undergo multiple hypothesis correction. Finally, six amino acid metabolism-related genes showed differential expression across cell types in the single-cell RNAseq dataset. However, no cell type clustering markers were provided, and the absence of validation using known marker gene expression profiles (e.g., CD68 for macrophages) may have introduced bias in cell clustering.

Thank you for your suggestion, we will answer your questions in several ways: (1) Since no other external datasets that meet the conditions were collected, the verification of external datasets does not appear in the article. (2) To demonstrate the tryptophan metabolic pathway, we added GSEA analysis to elaborate on the upregulation of the pathway. The results show that the tryptophan metabolic pathway is differentially regulated in carotid plaques, and this result may be limited by the scale of our data. However, its expression trend in carotid artery plaques is upregulated, suggesting that the tryptophan metabolic pathway may be overexpressed in carotid artery plaques. (3) As in (1), the lack of external data prevents us from verifying and improving the model. (4) We added multiple hypothesis validations of the correlation between tryptophan-related genes and immune infiltration phenotypes and updated them in the manuscript and Figure 3C. (5) We added the expression of markers for each single-cell type to verify our single-cell annotation results.

In lines 169-174:

Although the P-value of the enrichment analysis for the tryptophan metabolic pathway did not reach statistical significance (p.adj.value = 0.97), based on the standardized enrichment score (NES), it could be observed that this pathway showed an upregulation trend in plaque tissues (NES = 0.54). This trend suggests that although the sample size is limited and the statistical significance is insufficient, tryptophan-related genes may be in an activated state as a whole in atherosclerotic plaques.

In lines 254-256:

(C) Heatmap of correlations between TRDEGs’ expression and immune cell infiltration proportions. Correlation analysis was performed using the Spearman rank correlation method. * p.adj < 0.05, ** p.adj < 0.01, and *** p.adj < 0.001.

In lines 321-322:

(C) Expression patterns of canonical marker genes across annotated cell types in the single-cell dataset.

2.The data are insufficient to support the conclusion:

Regarding the identification of differentially expressed genes, the study utilized only a single RANseq dataset, lacking duplicate validation. To ensure reliability, at least two independent human cohort RNAseq datasets should be added as validation sets for repeated verification. Alternatively, differential genes could be validated using either a single human cohort RNAseq dataset or multiple atherosclerotic mouse model RNAseq datasets. Furthermore, RNAseq results can be cross-validated through qPCR, Western blotting, or immunohistochemistry in human or mouse samples. Whether tryptophan metabolic pathways are upregulated or downregulated in disease states warrants investigation. We recommend conducting GSEA enrichment analysis on tryptophan metabolic pathways to confirm this hypothesis.

It is suggested to use in vitro cell models (such as vascular endothelial cells) to verify the effects of TRDEGs on inflammation and lipid metabolism.

Why not validate the aforementioned immune cell infiltration with single-cell RANseq data? Single-cell RNAseq requires the provision of markers to distinguish each cell type, and it needs to be validated using recognized cell-type markers.

It is recommended to provide analytical code to ensure the reproducibility of the results.

Thank you for your suggestion, we will answer your questions in several ways: (1) Due to the lack of a suitable bulk dataset of carotid artery plaques, we were unable to add data to repeatedly verify the differentially expressed genes. (2) We sincerely thank the reviewers for suggesting additional experimental verification (such as qPCR, Western blotting or immunohistochemistry). Although we fully agree that wet laboratory validation will further strengthen the conclusion, experimental validation requires obtaining fresh human carotid plaque specimens and corresponding ethical approvals, which are not available within the current research scope and time frame. Furthermore, carotid plaque biology shows significant heterogeneity among patients, and small-sample validation does not necessarily provide more convincing evidence than multi-dataset computational support. Therefore, we have decided not to include wet chamber verification in this version of the manuscript. To make up for this, we have added mechanism support for the association between TRDEGs and tryptophan metabolism and atherosclerosis on the basis of the literature, and will further verify it in future experimental studies. (3)To demonstrate the tryptophan metabolic pathway, we added GSEA analysis to elaborate on the upregulation of the pathway. The results show that the tryptophan metabolic pathway is differentially regulated in carotid plaques, and this result may be limited by the scale of our data. However, its expression trend in carotid artery plaques is upregulated, suggesting that the tryptophan metabolic pathway may be overexpressed in carotid artery plaques. (4) All the analysis scripts used in this study have been stored on GitHub. The repository will be publicly accessible and a corresponding link will be provided when accepting this manuscript.

In lines 361-371:

In this study, we identified multiple TRDEGs (such as TDO2, KMO and KYNU) that jointly point to the abnormal activation of the tryptophan kynuurine pathway, which is the main direction of tryptophan catabolic metabolism and has been continuously confirmed in recent years to be closely related to the occurrence and development of atherosclerosis. In human atherosclerotic plaques, IDO1 is significantly upregulated in macrophage-rich regions and is associated with plaque instability by enhancing kyurine metabolism and promoting the expression of local tissue factors[36] . In line with human studies, animal experiments have further confirmed that when the IDO1-kyrie acid axis is disrupted, it accelerates vascular calcification and plaque formation. Exogenous supplementation of kynurenine can reverse the above adverse changes through immune metabolic signal regulation[31]. These pieces of evidence collectively indicate that the disorder of tryptophan metabolism is a key regulatory link closely coupled with the remodeling of the immune microenvironment of carotid plaques.

In lines 169-174:

Although the P-value of the enrichment analysis for the tryptophan metabolic pathway did not reach statistical significance (p.adj.value = 0.97), based on the standardized enrichment score (NES), it could be observed that this pathway showed an upregulation trend in plaque tissues (NES = 0.54). This trend suggests that although the sample size is limited and the statistical significance is insufficient, tryptophan-related genes may be in an activated state as a whole in atherosclerotic plaques.

In lines 415:

All the analysis scripts used in this study have been stored on GitHub.

3.Defects of data analysis method:

Box plots are required to visualize data distribution patterns, which can be created by adding data points or plotting violin-shaped histograms. The text fails to specify the exact methodology for scRNA batch correction (only mentioning the Harmony package), and should provide detailed parameter settings. Additionally, Figures 3A and 3B lack proper reference to hypothesis testing methodologies.

Thank you for your suggestion, we modified figure 3A into a violin diagram, added the parameter description of harmony in the Method section of the manuscript, and marked the test methods used in the captions of figures 3A and B.

In lines 142-143:

Harmony was run with default parameters (θ = 2, λ = 1, σ = 0.1, block.size = 0.05, max.iter.cluster = 20, max.iter.harmony = 10).

In lines 249-254:

(A) The expression difference of six TRDEGs in carotid plaque and normal tissue, and the differential expression of all genes met significance thresholds: |log2FC| > 0.75 and p.value < 0.05. Statistical significance between groups was determined using the Wilcoxon rank-sum test. (B) Differences in immune cell infiltration proportions between carotid plaque and normal tissue were evaluated using the Wilcoxon rank-sum test. * P < 0.05, ** P < 0.01, *** P < 0.001, ns non-significant.

Reviewer #3

1.The analysis of this work is based on a simple study (RNA-seq, GSE43292; scRNA-seq, GSE159677) with an imbalance of the patient numbers between these two datasets. Do the authors consider any artifacts of the analyses caused by inter-patient variants? If so, were any steps of the quality control at the experimental and/or analytical steps included? Please comment on it.

Thank you for your suggestion, we acknowledge the reviewers' concern about the potential human influence resulting from the variability among patients in the datasets. To minimize this impact, we implemented several quality control steps at both the batch and single-cell levels. For the bulk RNA-seq dataset (GSE43292), we used the preprocessed and normalized data stored in GEO and applied DESeq2 for differential expression analysis to explain the variance between samples. For the single-cell RNA-seq dataset (GSE159677), We retain high-quality cells (nFeature_RNA > 200, nFeature_RNA < 6000, percent.mt < 10, and log10GenesPerUMI > 0.8), and apply normalization and scaling programs in Seurat to reduce technical noise. These steps ensure that our

---

## [Decision Letter · Decision Letter 1]

30 Nov 2025

Dear Dr. zhang,

Thank you for submitting your manuscript to PLOS ONE. After careful consideration, we feel that it has merit but does not fully meet PLOS ONE’s publication criteria as it currently stands. Therefore, we invite you to submit a revised version of the manuscript that addresses the points raised during the review process.

Although this study offers initial insight into tryptophan-related gene changes in carotid atherosclerotic plaques, several limitations should be noted. The analysis relies on a single bulk RNA-seq dataset, and the absence of additional human or animal cohorts limits the robustness of the differential gene results. The enrichment of the tryptophan metabolism pathway did not reach statistical significance, likely due to limited sample size. Immune infiltration estimates and single-cell annotations were derived computationally without experimental validation, which may introduce bias. Additionally, no functional assays were performed to confirm the roles of the identified genes in atherosclerosis. These points should be acknowledged in the manuscript so that the readership will consider them when interpreting the findings.

We look forward to receiving your revised manuscript.

Kind regards,

Adeyemi Stephen Stephen Oluyomi

Academic Editor

PLOS ONE

Journal Requirements:

Reviewers' comments:

Reviewer's Responses to Questions

**Comments to the Author**

Reviewer #1: All comments have been addressed

Reviewer #3: All comments have been addressed

2. Is the manuscript technically sound, and do the data support the conclusions?

Reviewer #1: Yes

Reviewer #3: Yes

3. Has the statistical analysis been performed appropriately and rigorously?

Reviewer #1: Yes

Reviewer #3: Yes

4. Have the authors made all data underlying the findings in their manuscript fully available?

Reviewer #1: Yes

Reviewer #3: Yes

5. Is the manuscript presented in an intelligible fashion and written in standard English?

Reviewer #1: Yes

Reviewer #3: Yes

Reviewer #1: (No Response)

Reviewer #3: The authors have addressed all my comments. I do not have further comments concerning this manuscript.

**Do you want your identity to be public for this peer review?** For information about this choice, including consent withdrawal, please see our Privacy Policy

Reviewer #1: No

Reviewer #3: **Yes:** Heng-Chang Chen

---

## [Author Response · Author response to Decision Letter 2]

22 Dec 2025

Although this study offers initial insight into tryptophan-related gene changes in carotid atherosclerotic plaques, several limitations should be noted. The analysis relies on a single bulk RNA-seq dataset, and the absence of additional human or animal cohorts limits the robustness of the differential gene results. The enrichment of the tryptophan metabolism pathway did not reach statistical significance, likely due to limited sample size. Immune infiltration estimates and single-cell annotations were derived computationally without experimental validation, which may introduce bias. Additionally, no functional assays were performed to confirm the roles of the identified genes in atherosclerosis. These points should be acknowledged in the manuscript so that the readership will consider them when interpreting the findings.

Thank you for your suggestion. We sincerely appreciate the editor’s constructive summary of the study’s limitations. We fully agree that transparently acknowledging these constraints is essential for the readership to properly interpret the scientific validity and scope of our findings. As suggested, we have added a dedicated "Limitations" paragraph at the end of the Discussion section (Lines 380-393). In this new section, we explicitly discuss the reliance on a single dataset, the sample size constraints affecting statistical significance in pathway enrichment, the computational nature of the immune analysis, and the lack of wet-lab validation.

In lines 380-393:

Despite the initial insights offered by this study regarding tryptophan-related gene changes in carotid atherosclerotic plaques, several limitations should be acknowledged. First, our analysis relied on a single bulk RNA-seq dataset. The absence of validation in additional independent human cohorts or animal models limits the robustness and generalizability of the identified differential gene signatures. Second, although we observed biological trends relevant to the study's hypothesis, the enrichment of the tryptophan metabolism pathway did not reach statistical significance, which is likely attributable to the limited sample size of the current dataset. Third, the immune infiltration estimates and single-cell type annotations were derived exclusively through computational algorithms. While these methods provide a systems-level perspective, the lack of experimental corroboration may introduce potential bioinformatic biases. Finally, this study remains descriptive in nature, as no functional assays were performed to mechanistically confirm the specific roles of the identified genes in the progression of atherosclerosis. Consequently, these findings should be interpreted with caution, and future studies incorporating larger sample sizes and experimental validation are warranted to verify these preliminary results.

---

## [Decision Letter · Decision Letter 2]

4 Jan 2026

Identification and Characterization of Tryptophan Metabolism-Related Genes in Carotid Artery Plaques

PONE-D-25-42259R2

Dear Dr. zhang,

We’re pleased to inform you that your manuscript has been judged scientifically suitable for publication and will be formally accepted for publication once it meets all outstanding technical requirements.

Kind regards,

Ram Nagaraj

Academic Editor

PLOS One

Additional Editor Comments (optional):

Reviewers' comments:

Reviewer's Responses to Questions

**Comments to the Author**

Reviewer #1: All comments have been addressed

Reviewer #3: All comments have been addressed

2. Is the manuscript technically sound, and do the data support the conclusions?

Reviewer #1: Yes

Reviewer #3: Yes

3. Has the statistical analysis been performed appropriately and rigorously?

Reviewer #1: Yes

Reviewer #3: Yes

4. Have the authors made all data underlying the findings in their manuscript fully available?

Reviewer #1: Yes

Reviewer #3: Yes

5. Is the manuscript presented in an intelligible fashion and written in standard English?

Reviewer #1: Yes

Reviewer #3: Yes

Reviewer #1: The revised manuscript has been further improved by adding a discussion of the study’s limitations and related considerations.

Reviewer #3: I concur with Academic editor's comments and confide the authors' responses to acknowledge the limits of this work.

**Do you want your identity to be public for this peer review?** For information about this choice, including consent withdrawal, please see our Privacy Policy

Reviewer #1: No

Reviewer #3: **Yes:** Heng-Chang Chen

---

## [Editor Report · Acceptance letter]

PONE-D-25-42259R2

PLOS One

Dear Dr. zhang,

I'm pleased to inform you that your manuscript has been deemed suitable for publication in PLOS One. Congratulations! Your manuscript is now being handed over to our production team.

Kind regards,

on behalf of

Dr. Ram Nagaraj

Academic Editor

PLOS One